# PRISM: Patch Diffusion with Dynamic Retrieval Augmented Guidance and Permutation Invariant Conditioning

**Shivam Pal**  *pshivam@cse.iitk.ac.in*
*Dept. of Computer Science and Engineering IIT Kanpur*

**Avideep Mukherjee**  *avideep.mukherjee@dolby.com*
*Dolby Laboratories Inc.*

**Vinay P. Namboodiri**  *vpn22@bath.ac.uk*
*Dept. of Computer Science University of Bath*

**Piyush Rai**  *piyush@cse.iitk.ac.in*
*Dept. of Computer Science and Engineering and Dept. of Intelligent Systems*
*IIT Kanpur*

**Reviewed on OpenReview:** *https://openreview.net/forum?id=ru712j5D2d*

## Abstract

Diffusion models have achieved state-of-the-art results in image generation but often require extensive computational resources and large-scale datasets, limiting their practicality in resource-constrained settings. To address these challenges, we introduce PRISM, a retrieval-guided, patch-based method that trains solely on image patches instead of full resolution images. PRISM achieves superior global coherence and outperforms patch-only baselines, even when trained on only a fraction of the data. For each training example, PRISM retrieves semantically related neighbors from a disjoint retrieval set using CLIP embeddings. It aggregates their unordered signals with a Set Transformer, ensuring permutation-invariant conditioning that captures higher-order relationships. A dynamic neighbor-annealing schedule optimizes the contextual guidance over time, leading to more coherent results. Experiments on unconditional image generation tasks using CIFAR-10, CelebA, ImageNet-100, and AFHQv2 datasets, along with ablation studies, validate our approach, demonstrating that retrieval-augmented, set-based conditioning closes the coherence gap in patch-only diffusion.

## 1 Introduction

Diffusion models (Ho et al., 2020) have emerged as a dominant force in generative modeling, achieving state-of-the-art results across a spectrum of tasks, including photorealistic image synthesis (Dhariwal & Nichol, 2021; Ho et al., 2022), creative text-to-image generation (Rombach et al., 2022), and high-fidelity audio synthesis. However, their remarkable success comes at a significant computational cost. Training these models traditionally involves operating directly on full-sized images, which requires massive U-Net backbones that consume substantial memory and processing power. Such resource-intensive requirements effectively render state-of-the-art research inaccessible to the broader academic community, creating a bottleneck that slows the overall progress of the field.

A promising strategy to mitigate these costs is to train models on smaller, more manageable image patches. The recent *Patch Diffusion* framework by Wang et al. (2023) has made strides in this direction, demonstrating a significant reduction in computational requirements. However, this approach relies on a critical compromise. To learn global context and generate coherent images, the model must still be trained on full-resolution images for approximately 50% of the training steps. This hybrid strategy, while beneficial, only partially solves the efficiency problem and underscores a fundamental challenge. Patch-based training inherently struggles to

capture the long-range dependencies necessary for global image consistency. This leaves a crucial research gap and poses a fundamental question. *Is it possible to achieve the full computational benefits of a patch-only training regime without sacrificing the global coherence that full-resolution data provides?*

A key reason patch-only training struggles is the loss of global context, which leads to incoherent image structures. To overcome this, we propose **PRISM**: *$P$atch Diffusion with Dynamic $R$etrieval-Augmented Guidance and Permutation-$I$nvariant $S$et based Conditioning for Image $M$odeling.* PRISM eliminates the need for full-resolution images during training by introducing a retrieval-augmented guidance (RAG) mechanism. We partition our dataset into a training set and a retrieval set. For each image in the training set, we find its $k$-nearest neighbors from the retrieval set using the $\ell_2$ distance between their pre-computed CLIP (Radford et al., 2021) embeddings. These neighbors pro-

vide the global contextual information that is otherwise missing, guiding the model to generate coherent images while training only on small patches. A crucial challenge is how to effectively combine the information from these $k$ neighbors. Since the order of retrieved neighbors is arbitrary, the conditioning mechanism must be permutation-invariant. PRISM addresses this with a set-based conditioning approach. Instead of simply averaging embeddings, we use a *Set Transformer* (Lee et al., 2019) to process the set of $k$ CLIP embeddings. The Set Transformer produces a single, powerful aggregated signal that captures the complex relationships between the neighbors. This aggregated signal is then used as a conditioning input for the diffusion model, which is trained on patches using the EDM-DDPM++ framework by Karras et al. (2022).

Figure 1: Results on ImageNet-100 at $64{\times}64$ resolution. Both Patch Diffusion and PRISM are trained under the identical patch-only setting using $16{\times}16$ and $32{\times}32$ patches (no full-resolution images). PRISM achieves significantly better generation quality.

Furthermore, we find that the optimal number of retrieved neighbors, $k$, is not static. We introduce a dynamic annealing schedule for $k$, starting with a high value and gradually decreasing it as training progresses. This strategy allows the model to first learn from a broad set of contextual cues to establish global coherence and then gradually refine its focus on more specific details guided by the most relevant neighbors. As shown in Figure 1, PRISM significantly outperforms standard Patch Diffusion in a strict patch-only setting, demonstrating superior generation quality without requiring full-resolution training.

In summary, our contributions are:

1. We introduce PRISM, a patch-based diffusion framework that achieves strong image coherence without using any full-resolution images during training, relying instead on a retrieval-augmented guidance mechanism, thus significantly reducing the training set size.

2. We propose a permutation-invariant, set-based conditioning method using a Set Transformer to effectively aggregate contextual information from multiple retrieved images into a single, rich conditioning vector.

3. We introduce a dynamic annealing schedule for the number of retrieved neighbors $(k)$, which improves training stability and the fidelity of the final generated images.

## 2 Related Works

There has been substantial research on deep generative models for diverse data types, including images, videos, text, and multimodal data. This work centers primarily on diffusion probabilistic models by Sohl-Dickstein et al. (2015) and their variations (Kong & Ping, 2021; San-Roman et al., 2021; Song et al., 2021a; Dhariwal & Nichol, 2021; Ho et al., 2020; Ronneberger et al., 2015; Song et al., 2021b), particularly those optimized for compact model sizes, which are most pertinent to our objectives. Many diffusion-based model

variants, such as score-based models, have been explored across different data types (Yang et al., 2023). Our focus is on designing diffusion models that are parameter-efficient by design, enabling efficient training. Three main strategies include optimizing representation (using probability flow ODEs), training on latent representations, and employing patch-based training. For instance, consistency models by Song et al. (2023) approximate the stochastic differential equation derived from the diffusion process, aiming to reduce the number of function evaluations at inference time, although substantial training time remains necessary. Latent diffusion models (LDM) by Rombach et al. (2022) perform diffusion in latent space rather than image space, theoretically allowing for a smaller latent space to minimize computational costs. However, due to encoder-decoder limitations, LDMs typically rely on a larger latent space, requiring extensive GPU hours for training.

Patch-based training offers an alternative. For example, Mukherjee et al. (2024) trains and generates images patch-by-patch instead of using full-sized images and uses RAG (Blattmann et al., 2022) to compensate for the coherence problem. Wang et al. (2023) demonstrates that training on patches can reduce training time. Their approach, which involves training on varied patch sizes and periodically including full-sized images, does not yield reductions in model size. Another patch-based approach by Ding et al. (2024) incorporates a feature collage of patches to condition image generation.

Some recent work has started to look at how to adapt diffusion models with just a few examples (Lu et al., 2023). For example, DreamBooth by Ruiz et al. (2023) fine-tunes a pre-trained model using only four images of a person, adding a special prompt to help the model generate different images with the same person. Zhang et al. (2022) showed that just one image is enough to fine-tune a diffusion model, and the model can still generate high-quality images based on different prompts. However, these methods focus on improving pre-trained models, not training a model from scratch.

Our work more explicitly targets using patches only and getting rid of full-sized images during training to improve model efficiency. The consequent problem of incoherence is mitigated by conditioning the patches with relevant images retrieved from an external database, incorporating retrieval-based augmentation. This emphasis on coherence aligns with broader work in generative model coherence (Xu et al., 2018; Chu et al., 2020). We propose patch-based permutation-invariant retrieval-guided diffusion models, complementary to approaches such as probability flow ODE (Song et al., 2023) and latent diffusion (Rombach et al., 2022) and could be combined with them.

## 3 Background

Our method PRISM integrates three primary components, Diffusion modeling, Retrieval-Augmented Generation (RAG), and Set Transformers. In this section, we provide a brief overview of these components. Specifically, we cover (i) Denoising Diffusion Probabilistic Models (DDPMs), which serve as our generative model, (ii) retrieval mechanisms that enhance the model with non-parametric memory via external databases, and (iii) Set Transformers, utilized for their permutation-invariant properties to effectively aggregate retrieved neighbors into a dense conditioning representation.

### 3.1 Diffusion Models

Denoising Diffusion Probabilistic Models (DDPMs) (Ho et al., 2020; Sohl-Dickstein et al., 2015; Song et al., 2021b) are deep generative models built from a fixed forward noising process and a learnable reverse denoising process. The forward process is a Markov chain $q(x_{1:T} \mid x_0) = \prod_{t=1}^{T} q(x_t \mid x_{t-1})$ that gradually corrupts a clean sample $x_0$ into noise. In DDPMs each transition is Gaussian, $q(x_t \mid x_{t-1}) = \mathcal{N}(\sqrt{1 - \beta_t}\, x_{t-1},\, \beta_t \mathbf{I})$, with a variance schedule $\{\beta_t\}_{t=1}^{T}$. Writing $\alpha_t = 1 - \beta_t$ and $\bar{\alpha}_t = \prod_{s=1}^{t} \alpha_s$, the marginal at step $t$ has the closed form $q(x_t \mid x_0) = \mathcal{N}(\sqrt{\bar{\alpha}_t}\, x_0,\, (1 - \bar{\alpha}_t)\mathbf{I})$, or equivalently $x_t = \sqrt{\bar{\alpha}_t}\, x_0 + \sqrt{1 - \bar{\alpha}_t}\, \varepsilon$ with $\varepsilon \sim \mathcal{N}(0, \mathbf{I})$. For a suitable schedule, the marginal $q(x_T) = \int q(x_T \mid x_0)\, p_{\text{data}}(x_0)\, dx_0$ is close to $\mathcal{N}(0, \mathbf{I})$, so $x_T$ is approximately an isotropic Gaussian and largely independent of $x_0$. The closed-form expression for $q(x_t \mid x_0)$ allows direct sampling of noisy pairs $(x_t, x_0)$ at arbitrary timesteps without explicitly simulating the full forward chain.

The generative model defines a reverse Markov chain $p_\theta(x_{0:T}) = p(x_T) \prod_{t=1}^{T} p_\theta(x_{t-1} \mid x_t)$, with prior $p(x_T) = \mathcal{N}(0, \mathbf{I})$ and Gaussian transitions $p_\theta(x_{t-1} \mid x_t) = \mathcal{N}(\mu_\theta(x_t, t), \Sigma_\theta(x_t, t))$ parameterized by a neural network.

The ideal reverse conditionals $q(x_{t-1} \mid x_t)$ induced by the forward process are intractable, so the neural network parameters $\theta$ are trained to make $p_\theta(x_{t-1} \mid x_t)$ approximate them for all $t$. In practice, one often fixes $\Sigma_\theta$ to a simple diagonal form and parameterizes the mean via noise prediction. A network $\varepsilon_\theta(x_t, t)$ predicts the noise $\varepsilon$ in the reparameterization $x_t = \sqrt{\bar{\alpha}_t}\, x_0 + \sqrt{1 - \bar{\alpha}_t}\, \varepsilon$, from which $\mu_\theta(x_t, t)$ can be written in closed form. This leads to the widely used simplified training objective

$$\mathcal{L}(\theta) = \mathbb{E}_{t, x_0, \varepsilon}\big[\big\|\varepsilon - \varepsilon_\theta(x_t, t)\big\|_2^2\big]$$

where $t$ is sampled from a pre-defined distribution on $\{1, \ldots, T\}$, $x_0 \sim p_{\text{data}}$, and $\varepsilon \sim \mathcal{N}(0, \mathbf{I})$. Intuitively, the model learns, at each noise level, to denoise by predicting the perturbing noise vector.

After training, sampling proceeds by drawing $x_T \sim \mathcal{N}(0, \mathbf{I})$ and iteratively applying the learned reverse transitions $p_\theta(x_{t-1} \mid x_t)$ for $t = T, \ldots, 1$ to obtain $x_0$. At each step, the network either predicts the mean $\mu_\theta(x_t, t)$ or the noise $\varepsilon_\theta(x_t, t)$ and uses this to slightly denoise $x_t$, gradually steering the sample towards high-density regions of the data distribution. Unrolling these small denoising updates implements an iterative mapping from Gaussian noise to complex data distributions. In the limit of many small steps, this discrete reverse chain can be viewed as a time-discretized counterpart of score-based generative modeling, where the network implicitly captures the gradients $\nabla_{x_t} \log q(x_t)$ of the noised marginals (Song et al., 2021b).

## 3.2 Retrieval-Augmented Generation

An alternative strategy for creating compact deep generative models is Retrieval-Augmented Generation (RAG) Zhao et al. (2024), which augments a parametric generator with access to an external database $\mathcal{D}$. Instead of encoding all information about the data distribution in its parameters, the model can query $\mathcal{D}$ at training and inference time and retrieve a small, relevant subset of examples. Concretely, a retrieval function maps a query (e.g., the current input, conditioning signal, or a partially generated sample) to a subset $R(x) \subseteq \mathcal{D}$ of similar items, and the generator learns $p_\theta(x \mid c, R(x))$ rather than $p_\theta(x \mid c)$, where $c$ denotes any additional conditioning. This effectively equips the model with a non-parametric memory. The weights capture generalizable structure, while $\mathcal{D}$ provides fine grained, instance level information. Retrieval is typically performed in a learned embedding space, letting the model leverage local neighborhoods in $\mathcal{D}$ to improve fidelity and diversity without inflating the parameter count.

In recent work, Blattmann et al. (2022) applied RAG concepts to diffusion models using a non-parametric retrieval function $\xi_k : \mathcal{D} \to \mathcal{M}_{\mathcal{D}}^{(k)}$, where $\mathcal{M}_{\mathcal{D}}^{(k)}(x) \subseteq \mathcal{D}$ denotes the set of $k$ nearest neighbors of a sample $x \sim p(x)$. For each training sample, their retrieval-augmented diffusion model (RDM) computes $\mathcal{M}_{\mathcal{D}}^{(k)}(x)$ and conditions the noise-prediction network on representations of these neighbors, for example by concatenating features or using cross-attention over $\mathcal{M}_{\mathcal{D}}^{(k)}(x)$. This allows the model to focus on modeling residual variability around retrieved examples rather than synthesizing entire regions of the data space from scratch. The same retrieval mechanism is used at generation time, so sampling proceeds by repeatedly denoising while being guided by local neighborhoods in $\mathcal{D}$. Besides enabling more compact parameterizations, this retrieval-augmented formulation has been shown to improve global consistency and coherence across generated blocks in block-wise generative tasks, where standard diffusion models without retrieval often struggle to maintain long-range structure.

## 3.3 Set Transformers

Set Transformers (Lee et al., 2019) model unordered sets using permutation-equivariant attention blocks together with permutation-invariant pooling. We represent a set of $n$ elements as a matrix $\mathcal{X} \in \mathbb{R}^{n \times d}$, where each row is an element. Given $\mathcal{X} \in \mathbb{R}^{n \times d}$ and $\mathcal{Y} \in \mathbb{R}^{m \times d}$, scaled dot-product cross-attention is $\text{Att}(\mathcal{X}, \mathcal{Y}) = \text{softmax}\big((\mathcal{X}\mathbf{W}^q)(\mathcal{Y}\mathbf{W}^k)^\top / \sqrt{d_k}\big)\mathcal{Y}\mathbf{W}^v \in \mathbb{R}^{n \times d_v}$, with learnable $\mathbf{W}^q, \mathbf{W}^k \in \mathbb{R}^{d \times d_k}$ and $\mathbf{W}^v \in \mathbb{R}^{d \times d_v}$. Multi-head attention is obtained by computing several such attentions in parallel and linearly projecting the concatenated outputs, which we write as $\text{Multihead}(\mathcal{X}, \mathcal{Y}, \mathcal{Y})$.

The Multihead Attention Block (MAB) of Lee et al. (2019) is $H = \text{LN}(\mathcal{X} + \text{Multihead}(\mathcal{X}, \mathcal{Y}, \mathcal{Y}))$, $\text{MAB}(\mathcal{X}, \mathcal{Y}) = \text{LN}(H + \text{rFF}(H))$, where LN is layer normalization and rFF is a row-wise feed-forward

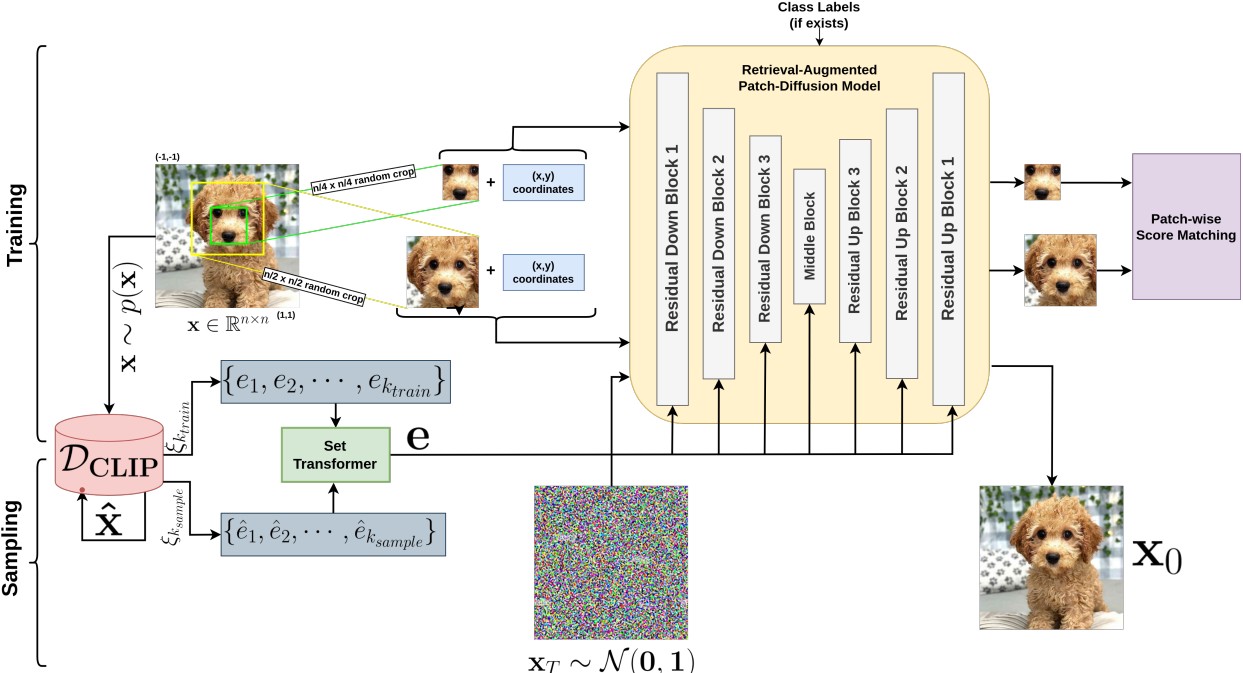

Figure 2: The training pipeline of PRISM involves creating a CLIP embedding database, selecting random image patches conditioned with retrieved neighbors via a retrieval strategy, processing these through a Set Transformer to generate conditioning representations, and feeding them into an EDM-DDPM++ (Karras et al., 2022) U-Net architecture (Ronneberger et al., 2015). In contrast, the sampling process follows similarly but uses randomly selected pseudo-queries from the CLIP database instead of training images.

network. Self-attention is then $\text{SAB}(\mathcal{X}) := \text{MAB}(\mathcal{X}, \mathcal{X})$. Since all operations are applied row-wise with shared parameters, MAB and SAB are permutation-equivariant in $\mathcal{X}$.

A key innovation of Set Transformers is their use of the Induced Set Attention Block (ISAB) to improve computational efficiency. The ISAB introduces a set of $m$ inducing points, denoted as $\mathcal{I} = \{i_1, i_2, \ldots, i_m\} \in \mathbb{R}^{m \times d}$, where $m$ is much smaller than $n$ (i.e., $m \ll n$). $\text{ISAB}(\mathcal{X}) := \text{MAB}(\mathcal{X}, \text{MAB}(\mathcal{I}, \mathcal{X})) \in \mathbb{R}^{n \times d}$. It reduces the overall complexity from $O(n^2)$ to $O(nm)$. This design allows Set Transformers to scale effectively with larger datasets by mitigating the quadratic growth in computation typically associated with self-attention mechanisms.

## 4    Method

Figure 2 describes the training and sampling procedure of PRISM. First, we construct a retrieval set $\mathcal{D}_{\text{CLIP}}$ containing CLIP embeddings of all full training images. This database is used exclusively for retrieval and is never passed directly to the diffusion backbone. During training, we sample a full training image $x$ and use its *full-image* CLIP embedding as a query to retrieve $\xi_k$ nearest neighbors from $\mathcal{D}_{\text{CLIP}}$, where $\xi_k$ is determined by a dynamic scheduling strategy. In parallel, we extract a random patch $x^{\text{patch}}$ from $x$, and only this patch is used as the input to the U-Net denoising network (i.e., the EDM-DDPM++ $\epsilon_\theta$ parameterized by $\theta$), not the full image. The retrieved neighbors are processed by a Set Transformer to produce a final conditioning representation, which is fed into $\epsilon_\theta$ alongside the noisy patch. The sampling process is similar to the training process, except instead of using an actual training image, a pseudo-query is selected from $\mathcal{D}_{\text{CLIP}}$ to condition the model. The pseudo-query serves as a proxy input during sampling, allowing the model to generate or sample new images based on the learned conditional structure from the training phase.

## 4.1 Creating the CLIP Database

To build the CLIP database for guiding the model, we start with a dataset denoted as $\mathcal{X}$. We randomly partition $\mathcal{X}$ into a retrieval subset $\mathcal{D} \subseteq \mathcal{X}$ and a training subset $\mathcal{X}_{\text{train}} = \mathcal{X} \setminus \mathcal{D}$. When class labels are available, this partition is performed in a class-wise (stratified) manner so that the class distribution is approximately preserved in both $\mathcal{D}$ and $\mathcal{X}_{\text{train}}$. In all our experiments, we choose the retrieval pool to be larger than the training set, i.e., $|\mathcal{D}| > |\mathcal{X}_{\text{train}}|$, to ensure a diverse and rich set of neighbors for retrieval. The subset $\mathcal{D}$ is used to create the retrieval set $\mathcal{D}_{\text{CLIP}}$, while $\mathcal{X}_{\text{train}}$ is used to train the diffusion model.

Specifically, $\mathcal{D}_{\text{CLIP}} = \{e_i\}_{i=1}^n$, where $e_i$ is the CLIP embedding of image $x_i \in \mathcal{D}$ and $n = |\mathcal{D}|$ is the total number of images in the retrieval subset. CLIP embeddings are generated using a pre-trained CLIP model, which projects images and textual descriptions into a shared embedding space. In this work, we use the ViT-B/32 (Vision Transformer) backbone (Dosovitskiy et al., 2021) as our CLIP encoder, which produces 512-dimensional embeddings for each image.

## 4.2 Extraction of Neighbors and Patches

For each training image $x \in \mathcal{X}_{\text{train}}$, we first use the *full-resolution* image to retrieve neighbors from the CLIP database $\mathcal{D}_{\text{CLIP}}$. Concretely, a retrieval strategy $\xi_{k_{\text{train}}}$ (with $k_{\text{train}}$ given by the dynamic nearest-neighbor scheduling in Section 4.4) returns the set of CLIP neighbors $\mathcal{M}_{\mathcal{D}_{\text{CLIP}}}^{(k_{\text{train}})}(x) \coloneqq \{e_1, e_2, \ldots, e_{k_{\text{train}}}\}$, where each $e_i$ is the CLIP embedding of a retrieved image from $\mathcal{D}$.

After retrieval, we operate purely at the patch level. Following the stochastic patch-size scheduling principle of Wang et al. (2023), but without ever using the full image as input to the U-Net, we sample a patch with resolution $s$ from

$$s = \begin{cases} R/2 & \text{with probability } 3/5, \\ R/4 & \text{with probability } 2/5. \end{cases} \tag{1}$$

where $R$ denotes the resolution of the full image. Given the sampled $s$, we crop a patch from $x$ at random spatial coordinates $(i, j)$ and augment this patch by concatenating the $i$ and $j$ coordinate maps as two additional channels. The resulting coordinate-augmented patch is then used as the input to the denoising U-Net, while the retrieved neighbors $\mathcal{M}_{\mathcal{D}_{\text{CLIP}}}^{(k_{\text{train}})}(x)$ provide the conditioning signal via the Set Transformer.

## 4.3 Conditioning via Set Transformers

For each training image $x \in \mathcal{X}_{\text{train}}$, the retrieval mechanism yields a variable-sized set of CLIP embeddings $\mathcal{M}_{\mathcal{D}_{\text{CLIP}}}^{(k_{\text{train}})}(x) = \{e_1, \ldots, e_{k_{\text{train}}}\}$ (Section 4.2). To utilize this set as a global prior for the patch-level diffusion process, we must map these $k_{\text{train}}$ vectors into a single, fixed-dimensional conditioning vector $\mathbf{e}$. Crucially, because $k_{\text{train}}$ changes continuously under our dynamic annealing schedule, trivial aggregation strategies such as concatenation are architecturally incompatible, as they require a fixed input dimension. Conversely, simple averaging or processing the embeddings through a standard transformer followed by mean pooling (similar to Deep Sets (Zaheer et al., 2017)), discards higher-order structural interactions among the retrieved neighbors. To resolve this, we process the neighbor set using a Set Transformer (Lee et al., 2019). The Set Transformer employs permutation-equivariant self-attention blocks (SAB) followed by Pooling by Multihead Attention (PMA). This yields an order-agnostic, fixed-dimensional summary $\mathbf{e}$ that explicitly models the non-linear relationships within the neighbor set. By enforcing permutation invariance, the conditioning mechanism is robust to minor stochasticity in the $\ell_2$ retrieval ranking. This prevents the model from overfitting to a specific retrieval ordering and forces it to rely on the generalized semantic structure of the latent neighborhood.
The resulting global conditioning vector $\mathbf{e}$ is then injected into the U-Net backbone alongside the timestep embedding to guide the local patch denoising process. We employ the Set Transformer as a natural choice for permutation-invariant set aggregation. However, alternative sequence to vector projection architectures, such as the Q-Former (Li et al., 2023), could also be used.

Let $x_t^{i,j,s}$ denote the noised patch at time step $t$ obtained from $x$ by cropping at coordinates $(i, j)$ with patch size $s$ (Section 4.2), and let $\varepsilon \sim \mathcal{N}(0, \mathbf{I})$ be the injected noise. The U-Net $\varepsilon_\theta$ of the EDM-DDPM++

backbone is conditioned on the patch, timestep, and neighbor embedding, and is trained with the standard noise-prediction objective

$$\min_\theta \mathcal{L}(\theta) = \mathbb{E}_{x,\,t,\,i,\,j,\,s,\,\varepsilon}\big[\big\|\varepsilon - \varepsilon_\theta(x_t^{i,j,s}, t, \mathbf{e})\big\|_2^2\big], \tag{2}$$

where $x \sim \mathcal{X}_{\text{train}}$, $t$ is sampled from the diffusion timestep schedule, and $\mathbf{e}$ is the Set Transformer embedding of $\mathcal{M}_{\mathcal{D}_{\text{CLIP}}}^{(k_{\text{train}})}(x)$.

### 4.4 Dynamic Nearest Neighbor Scheduling

While optimizing Equation 2, we treat $k_{\text{train}}$ as a function of training progress rather than a fixed constant. Empirically, we find that a *decreasing* schedule for $k_{\text{train}}$ (starting from a larger value $k_{\text{max}}$ and gradually annealing to 1) significantly outperforms fixed, increasing, or randomly sampled schedules, leading to faster convergence and improved final performance. A detailed comparison of different scheduling strategies is provided in Section 6.2. A plausible explanation for the effectiveness of a decreasing $k_{\text{train}}$ schedule is given by a curriculum-learning perspective. Early in training, conditioning the Set Transformer on many neighbors yields a smoother, more global context embedding that captures broad structure in the dataset and stabilizes optimization. As training progresses and $k_{\text{train}}$ decreases, the conditioning becomes more focused, emphasizing a smaller set of closer neighbors and encouraging the model to learn finer-grained, instance-specific features. This coarse-to-fine progression mirrors curriculum learning, where the model is first exposed to easier, more averaged information and then gradually shifted toward more specific, detailed signals.

We define the dynamic annealing schedule as follow, let $E_{\text{tot}}$ denote the total number of training epochs and $k_{\text{max}}$ the initial number of neighbors. For clarity, we assume $E_{\text{tot}}$ is divisible by $k_{\text{max}}$ and define the number of epochs per $k$-value as $E_{\text{seg}} = \frac{E_{\text{tot}}}{k_{\text{max}}}$. Then, for epoch $e \in \{1, \ldots, E_{\text{tot}}\}$, we define

$$k_{\text{train}}(e) = k_{\text{max}} + 1 - \lceil e/E_{\text{seg}} \rceil, \tag{3}$$

which yields a piecewise-constant, monotonically decreasing schedule that starts from $k_{\text{train}}(1) = k_{\text{max}}$ and ends at $k_{\text{train}}(E_{\text{tot}}) = 1$, with each integer value of $k_{\text{train}}$ used for exactly $E_{\text{seg}}$ epochs.

While the retrieval set $\mathcal{D}$ and the training set $\mathcal{X}_{\text{train}}$ are disjoint, semantic overlap between them naturally exists and serves as the intended global semantic prior. Importantly, this overlap does not cause information leakage or trivial memorization, whereby the network utilizes the conditioning vector as a unique identifier to overfit the training data. PRISM structurally prevents this deterministic mapping through two mechanisms. First, the diffusion process operates on patches extracted at stochastic spatial coordinates and scales, requiring the model to learn local texture manifolds rather than rigid global coordinate lookup. Second, the conditioning vector is generated via a permutation-invariant Set Transformer operating over $k_{\text{train}}$ retrieved neighbors. During the critical early phases of the dynamic training schedule, the network is conditioned on an aggregated vector that represents a smoothed distributional semantic neighborhood rather than a unique image instance. This bottleneck destroys instance-specific identifiers, forcing the model to learn a generalizable generative mapping from the abstract semantic prior to the local patch distribution.

### 4.5 Sampling

During sampling, we first select a pseudo-query image $\hat{x}$ from the retrieval subset $\mathcal{D}$ and compute its CLIP embedding $\hat{e} = e(\hat{x}) \in \mathcal{D}_{\text{CLIP}}$. Using $\hat{e}$ as a query in CLIP space, we retrieve a neighbor set $\mathcal{M}_{\mathcal{D}_{\text{CLIP}}}^{(k_{\text{sample}})}(\hat{x}) := \{e_1, e_2, \ldots, e_{k_{\text{sample}}}\}$ from $\mathcal{D}_{\text{CLIP}}$ via the retrieval strategy $\xi_{k_{\text{sample}}}$. In principle, $k_{\text{sample}}$ could be tuned analogously to $k_{\text{train}}$ (Section 4.4) however, we observe that $k_{\text{sample}} = 1$ consistently gives the better results on all the datasets except high resolution (256×256). Detailed ablation study on $k_{\text{sample}}$ is given in section 8.

As in training, the retrieved neighbor embeddings are passed through the Set Transformer (Section 4.3) to obtain a conditioning embedding $\mathbf{e}$, which is provided to the U-Net at each denoising step. In contrast to the training procedure, PRISM generates *full-resolution* images at test time, without extracting patches or blocks. Following the strategy of Wang et al. (2023), we exploit the fully convolutional nature of the U-Net. Although the network is trained on patches, it can be directly applied to full-sized images by starting from

| Dataset | Method | Model Size | Retrieval Set Size | Training Set Size | Traning Time | FID Score |
|---|---|---|---|---|---|---|
| CIFAR-10 32x32 | Patch Diffusion (Wang et al., 2023) | 11M | - | 50K | $\sim$ 1.5 hrs | 42.59 |
| | RISSOLE (Mukherjee et al., 2024) | 50M | - | 50K | $\sim$ 5 hrs | 28.77 |
| | PRISM | 13M | 40K | 10K | $\sim$ 2 hrs | **24.68** |
| CelebA 64x64 | Patch Diffusion | 35M | - | 232K | $\sim$ 10 hrs | 21.99 |
| | PRISM | 42M | 162K | 70K | $\sim$ 14 hrs | **16.46** |
| ImageNet-100 64x64 | Patch Diffusion | 63M | - | 130K | $\sim$ 50 hrs | 99.11 |
| | PRISM | 63M | 100K | 30K | $\sim$ 50 hrs | **46.30** |
| AFHQv2 64x64 | Patch Diffusion | 27M | - | 15K | $\sim$ 11 hrs | 14.47 |
| | PRISM | 13M | 10K | 5K | $\sim$ 10 hrs | **10.02** |

Table 1: FID score ($\downarrow$) for CelebA, CIFAR-10, ImageNet-100 and AFHQv2 dataset.

$x_T \sim \mathcal{N}(0, \mathbf{I})$ at the target resolution and running the reverse diffusion process conditioned on $\mathbf{e}$ to obtain $x_0$. Training on patches reduces memory and computation while encouraging the model to learn strong local representations, whereas sampling at full resolution avoids stitching artifacts and block-wise inconsistencies, leading to globally coherent generations.

## 5 Experiments

### 5.1 Datasets

We evaluate PRISM on four standard benchmarks: CIFAR-10 $32 \times 32$ (Krizhevsky et al., 2014), CelebA $64 \times 64$ (Liu et al., 2015), ImageNet-100 $64 \times 64$ (Chrabaszcz et al., 2017; Deng et al., 2009), and AFHQv2 (Choi et al., 2020) $64 \times 64$ and $256 \times 256$. CIFAR-10 contains 60,000 natural images from 10 object classes and serves as a low-resolution, multi-class benchmark. CelebA provides over 232,000 aligned face images, suitable for structured facial generation. ImageNet-100 is a 100-class subset of ImageNet-1K with roughly 130,000 images, testing performance on more diverse object categories at $64 \times 64$ resolution. AFHQv2 contains high-quality animal face images (dog, cat, wild), which we downsample to $64 \times 64$ to evaluate generation on a multi-species but still structured domain.

### 5.2 Model Architecture

In our experiments, we use the EDM-DDPM++ (Karras et al., 2022), a Unet-based diffusion model known for effective image generation. To achieve a compact architecture, we reduce both the number of U-Net blocks and the channel dimensions, balancing model size with quality. This streamlined version of EDM-DDPM++ is consistently applied across all datasets. Training hyperparameters and model configurations, including block numbers and channel dimensions, are detailed in the appendix. All models were trained on a single Nvidia RTX 24GB GPU, which provided sufficient memory and efficiency.

### 5.3 Comparison with Baselines

We evaluate generation quality using the Fréchet Inception Distance (FID), computed with 50,000 generated samples and all real training images of each dataset as the reference distribution (50K for CIFAR-10, 232K for CelebA, 130K for ImageNet-100, and 15K for AFHQv2). We compare PRISM against Patch Diffusion (Wang et al., 2023) on all datasets and against RISSOLE (Mukherjee et al., 2024) on CIFAR-10 (Table 1), reporting FID, model size, training set size, training time, and retrieval set size.

The Set Transformer used in PRISM adds roughly 2M parameters on CIFAR-10 and about 7M parameters on CelebA, while the backbone remains the Patch Diffusion architecture. On ImageNet-100 we match the 63M parameter backbone of Patch Diffusion, and on AFHQv2 we use a compact 13M parameter model (including

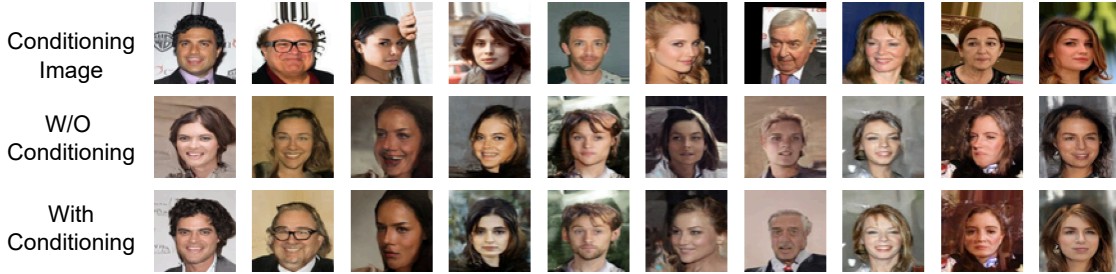

Figure 3: First row: real samples whose CLIP embeddings are used to extract the global conditioning signal via the Set Transformer. Second and third rows: images generated by PRISM without and with the global conditioning signal, respectively.

the Set Transformer), smaller than the 27M parameter Patch Diffusion baseline. Thus, the observed gains are not due to a radically larger generator but to retrieval-guided conditioning.

On CIFAR-10, PRISM attains an FID of 24.68, improving over Patch Diffusion (42.59) and RISSOLE (28.77), despite being trained on only 10K images versus 50K for both baselines. Training takes about 2 hours, slightly more than Patch Diffusion (∼1.5 hours) but far less than RISSOLE (∼5 hours). On CelebA, PRISM achieves an FID of 16.46 compared to Patch Diffusion's 21.99, using 70K training images instead of 232K, with training times of ∼14 hours and ∼10 hours respectively. On ImageNet-100, PRISM reduces FID from 99.11 (Patch Diffusion) to 46.30, while using only 30K training images compared to 130K, with similar training times (∼50 hours). Finally, on AFHQv2, PRISM improves FID from 14.47 to 10.02 while training on 5K images instead of 15K and with slightly shorter training time (10 vs. 11 hours). Overall, PRISM consistently achieves better FID scores while using substantially fewer training images, and with comparable or only moderately increased training time relative to Patch Diffusion. The Set Transformer–based conditioning over retrieved CLIP neighbors enables the model to exploit non-parametric information from the retrieval pool, leading to higher-quality and more data-efficient generation than purely parametric patch-based baselines.

### 5.4 Effectiveness in Limited-Data Regimes

In this section, we examine the performance of PRISM under limited-data settings. Table 2 reports FID scores when training on progressively smaller subsets of CIFAR-10, while keeping the evaluation protocol fixed (50,000 generated samples and the full CIFAR-10 training set as reference). With 10,000 training images, PRISM achieves an FID of 24.68, already improving over patch-based baselines trained on the full 50K set. When the training set is reduced to 5,000 images, the FID only slightly degrades to 25.67, and even with just 1,000 images, the FID remains competitive at 26.62. These results indicate that PRISM is able to maintain high sample quality even when trained on as little as 10% or 2% of the original data.

| Training set size | FID (↓) |
|---|---|
| 10K | 24.68 |
| 5K | 25.67 |
| 1K | 26.62 |

Table 2: FID on CIFAR-10.

We hypothesize that this robustness arises from the division of roles between the parametric model and the retrieval mechanism. The training subset is primarily used to learn how to map retrieved CLIP neighbors, aggregated by the Set Transformer, to patch-level denoising updates, while the retrieval database provides a large, fixed pool of examples that supplies rich non-parametric context. As long as there is sufficient data to learn the conditioning and denoising mappings, PRISM can leverage the retrieval pool to compensate for reduced training data. A natural concern in low-data regimes is overfitting or memorization. However, as illustrated in Figure 3, generated samples conditioned on a given real image do not simply replicate that image. Instead, PRISM performs a form of style transfer, outputs share high-level style and semantics with

the conditioning image while exhibiting distinct content and configurations, suggesting that the model has learned to generalize rather than memorize.

### 5.5 High-Resolution Experiments

To evaluate scalability, we train PRISM on the AFHQv2 dataset at $256 \times 256$ resolution and compare it to the Patch Diffusion. As shown in Table 3, PRISM significantly accelerates convergence and improves image quality using less training data and computation time. PRISM achieves superior FID scores with both CLIP (30.62) and DINOv2 (28.18) encoders in approximately 9 hours, utilizing only 5K training images. In contrast, the Patch Diffusion baseline requires 15K images and 20 hours to reach convergence (FID 36.43). When restricted to a comparable 10-hour budget, the baseline fails to converge (FID 57.80). While PRISM improves FID and Precision, it yields a lower Recall compared to the Patch Diffusion. This trade-off is expected in retrieval-augmented generation. Conditioning on retrieved neighbors constrains the generated outputs to regions well-represented by the retrieval set. This structural restriction prevents low-quality artifacts (increasing Precision) but limits the generation of rare, diverse samples (decreasing Recall). Visual samples generated by PRISM and Patch Diffusion are provided in Appendix Figures 4 and 5, respectively.

| Method | Model Size | Retrieval Set Size | Training Set Size | Training Time | FID ↓ | Precision ↑ | Recall ↑ |
|---|---|---|---|---|---|---|---|
| Patch Diffusion | 23M | - | 15K | $\sim$ 10 hrs | 57.80 | 0.226 | 0.086 |
| Patch Diffusion | 23M | - | 15K | $\sim$ 20 hrs | 36.43 | 0.335 | **0.172** |
| PRISM (CLIP) | 27M | 10K | 5K | $\sim$ 9 hrs | 30.62 | **0.382** | 0.124 |
| PRISM (DINOv2) | 27M | 10K | 5K | $\sim$ 9 hrs | **28.18** | 0.364 | 0.130 |

Table 3: Results on AFHQv2 $256 \times 256$ dataset.

## 6 Ablation Studies

### 6.1 Effectiveness of Each Component

Our method extends Patch Diffusion by incorporating a global conditioning signal derived from a retrieval-augmented pipeline. Given a training image from $\mathcal{X}_{\text{train}}$, we retrieve its $k$-nearest CLIP neighbors from the retrieval set $\mathcal{D}$ and aggregate their embeddings using a Set Transformer to obtain a global conditioning vector, which is then combined with the patch-level input. To quantify the contribution of each component, we start from the Patch Diffusion baseline (**PatchD**) and incrementally add retrieval (RAG), the Set Transformer (ST), and the dynamic neighbor scheduling (DS) described in Section 4.4.

**PatchD + RAG.** In this configuration, we augment Patch Diffusion with a simple retrieval-based global conditioning: for each training image, we retrieve its $k$ nearest neighbors in CLIP space from $\mathcal{D}_{\text{CLIP}}$ and obtain the global conditioning vector by averaging their embeddings. The value of $k$ is fixed during both training and sampling. As shown in Table 4, this naive RAG scheme already yields large gains: FID drops from 21.99 to 18.25 on CelebA and from 42.59 to 26.00 on CIFAR-10. Importantly, these improvements are achieved while training on $\mathcal{X}_{\text{train}}$ (a subset of $\mathcal{X}$), highlighting that even simple retrieval-based conditioning can substantially improve patch-based diffusion and make it more data-efficient.

**PatchD + RAG + ST.** Next, we replace the averaging operation with a Set Transformer that takes the $k$ neighbor embeddings as input and produces a learned global conditioning vector. The value of $k$ is still fixed. The Set Transformer can model non-linear interactions and dependencies between neighbors instead of treating them as an independent set, leading to a more expressive and robust global representation. This yields further improvements, indicating that better aggregation of retrieval information translates directly into better generation quality.

| Method | Trained on | CelebA | CIFAR-10 |
|---|:---:|:---:|:---:|
| PatchD | $\mathcal{X}$ | 21.99 | 42.59 |
| RAG + PatchD | $\mathcal{X}_{\text{train}}$ | 18.25 | 26.00 |
| ST + RAG + PatchD | $\mathcal{X}_{\text{train}}$ | 17.82 | 25.62 |
| ST + RAG + PatchD + DS (PRISM) | $\mathcal{X}_{\text{train}}$ | **16.46** | **24.68** |

Table 4: Ablation on CelebA and CIFAR-10: effect of retrieval (RAG), Set Transformer (ST), and dynamic scheduling (DS). PatchD uses the full dataset $\mathcal{X}$; all retrieval-based variants are trained only on $\mathcal{X}_{\text{train}}$.

**PatchD + RAG + ST + DS (PRISM).** Finally, our full model (PRISM) introduces dynamic scheduling (DS) of $k_{\text{train}}$ during training (Section 4.4). Rather than using a fixed number of neighbors, we start from a larger $k_{\text{max}}$ and gradually decrease to $k = 1$ over the course of training, allocating the same number of epochs to each $k$ value. This implements a coarse-to-fine curriculum, early on, the model sees broader neighborhoods (smoother global context), and later it focuses on fewer, closer neighbors (finer, instance-specific detail). This configuration achieves the best FID scores on both datasets , clearly outperforming the Patch Diffusion baseline trained on the full dataset.

Overall, Table 4 shows a consistent pattern. Each additional component (RAG, ST, DS) yields a measurable improvement, and the full PRISM model is strictly better than Patch Diffusion despite being trained on fewer images. This suggests that retrieval-guided, Set Transformer based conditioning is more effective at capturing global structure than relying solely on the patch-level diffusion backbone, and that dynamically controlling the neighborhood size further strengthens this effect.

## 6.2 Effect of Scheduling $k_{\text{train}}$

We study how different schedules for the number of training neighbors $k_{\text{train}}$ affect image quality. On CelebA, we compare four strategies: a linearly decreasing schedule, a linearly increasing schedule, a random schedule, and a fixed schedule. All models are trained on the same 1K-image subset with identical backbones, Set Transformer size, and total number of epochs; only the evolution of $k_{\text{train}}$ over training differs. In the decreasing schedule, training starts with a large neighborhood and gradually reduces $k_{\text{train}}$ down to 1, spending an equal number of epochs at each value. The increasing schedule does the opposite, starting from 1 and ramping up to a larger maximum. The random schedule samples a value of $k_{\text{train}}$ uniformly at each epoch, and the fixed schedule uses a single constant value throughout.

Table 6 reports FID for several choices of $k_{\text{sample}}$ at test time. Two patterns clearly emerge. First, for every $k_{\text{sample}}$, the decreasing schedule yields the best FID, often by a large margin compared to the other three strategies. Second, the lowest FID overall is achieved when the model is trained with a decreasing schedule and evaluated with $k_{\text{sample}} = 1$, i.e., with minimal conditioning at test time. This indicates that how we expose the Set Transformer to neighborhoods during training has a much stronger effect on performance than how many neighbors we provide at sampling.

A natural interpretation is curriculum learning. Early in training, a large $k_{\text{train}}$ provides a broad, stable context: the Set Transformer aggregates many neighbors and therefore learns smooth, global structure that is easier to fit when the model is still undertrained. As training progresses, a smaller $k_{\text{train}}$ forces the model to rely on fewer, closer neighbors, encouraging it to capture more fine-grained, instance-specific regularities. The decreasing schedule thus implements a coarse-to-fine curriculum over neighborhood size. In contrast, the increasing schedule starts with small, noisy neighborhoods when the model is weakest and only later introduces rich context, which appears less helpful. Random and fixed schedules do not follow any meaningful progression and consistently underperform.

Based on these observations, we adopt the decreasing schedule for all main experiments, combined with $k_{\text{sample}} = 1$ at test time. Concretely, we use a larger initial neighborhood size on CelebA, ImageNet-100, and AFHQv2 (starting from 32 neighbors) and a slightly smaller one on CIFAR-10 (starting from 16 neighbors), always annealing down to a single neighbor by the end of training.

## 6.3 Sensitivity to the Retrieval Encoder

To evaluate the framework's sensitivity to the choice of the pre-trained encoder, we ablate CLIP by substituting it with DINOv2 (Oquab et al., 2024). While CLIP is optimized for global semantic alignment, DINOv2 is trained to capture dense, localized structural correspondences. As shown in Table 1, evaluating on the AFHQv2 dataset at 256x256 resolution, replacing CLIP with DINOv2 improves the FID score from 30.62 to 28.18. This indicates that the permutation-invariant conditioning mechanism is robust to the specific latent topology of the encoder. PRISM actively leverages the finer-grained spatial priors provided by DINOv2 to better coordinate the local patch diffusion. Furthermore, because class labels are not explicitly provided during the retrieval step, semantic consistency between the generated output and the implicit query class depends on the clustering properties of the chosen latent space. Both CLIP and DINOv2 exhibit strong separability for object classes. Consequently, the $\ell_2$ nearest neighbors are predominantly class-consistent with the query image, mitigating the risk of out-of-class conditioning conflicts during generation.

## 7 Memory, Scalability, and Trade-offs

**Memory and Retrieval Overhead.** The memory footprint of the retrieval database $\mathcal{D}$ is heavily compressed by the encoder. The storage requirement scales as $\mathcal{O}(|\mathcal{D}| \cdot d)$ for latent dimension $d$, constituting a negligible footprint relative to raw pixel data. Because the nearest-neighbor lookups are deterministic given a static $\mathcal{D}$, they are computed entirely offline. The exact $\ell_2$ search complexity scales as $\mathcal{O}(|\mathcal{X}_{\text{train}}| \cdot |\mathcal{D}| \cdot d)$, adding zero latency to the U-Net training passes. For large-scale deployment, approximate nearest neighbor indexing reduces the per-query lookup complexity to $\mathcal{O}(\log |\mathcal{D}|)$, ensuring asymptotic scalability.

**Trade-offs versus Full-Resolution Training.** PRISM provides an efficient alternative to full-resolution training by decoupling model capacity from spatial resolution. This introduces three operational trade-offs:

- **Memory:** Standard diffusion attention scales spatially as $\mathcal{O}(R^4)$ for resolution $R \times R$. PRISM strictly bounds this to a fixed patch size ($\mathcal{O}(s^4)$ for $s \ll R$), eliminating the primary hardware bottleneck.

- **Sample Efficiency:** By offloading the global prior to a pre-trained encoder via RAG, PRISM converges using a fraction of the data and compute required by standard baselines.

- **Quality (Precision vs. Recall):** PRISM restricts synthesis to the localized manifold of $k$ retrieved neighbors. This suppresses structural artifacts (yielding higher Precision and FID) but inherently limits generative diversity to the semantic modes present in $\mathcal{D}$ (lowering Recall).

**Generation Diversity Trade-offs.** Retrieval-augmented conditioning can reduce generative diversity by biasing samples toward regions well represented in the retrieval set. We quantify this effect using Precision and Recall, where Precision reflects sample fidelity and Recall reflects manifold coverage. As shown in Table 3, PRISM (CLIP) improves Precision from 0.335 to 0.382 relative to the converged Patch Diffusion baseline, a 14% relative gain, while Recall decreases from 0.172 to 0.124, a 28% relative reduction. This behavior follows from the retrieval prior. Retrieved neighbors guide the reverse diffusion process toward image regions supported by $\mathcal{D}$, suppressing implausible structures and improving FID and Precision. However, rare or atypical samples weakly represented in $\mathcal{D}$ become less likely under this conditioning, lowering Recall. Encoder choice also affects this trade-off. PRISM with DINOv2 obtains slightly higher Recall than CLIP (0.130 vs. 0.124), consistent with DINOv2's patch-level training objective capturing finer-grained structural correspondences than CLIP's global image–text alignment. This suggests that structurally richer encoders and diversity-aware retrieval-set construction may help mitigate diversity loss.

## 8 Inference-Time Retrieval Size ($k_{\text{sample}}$)

We analyze the role of the Set Transformer at inference by varying the number of retrieved neighbors $k_{\text{sample}}$ while keeping the trained model fixed. This evaluates whether single-neighbor inference is sufficient, and whether multi-neighbor aggregation provides additional benefit at test time. Table 5 shows that the optimal

| Resolution | Encoder | $k_{\mathrm{sample}}$ | FID ↓ | Precision ↑ | Recall ↑ |
|---|---|---|---|---|---|
| $64 \times 64$ | CLIP | 1 | 10.62 | 0.615 | 0.486 |
| $64 \times 64$ | CLIP | 4 | 12.31 | 0.574 | 0.473 |
| $64 \times 64$ | CLIP | 8 | 12.67 | 0.579 | 0.458 |
| $256 \times 256$ | CLIP | 1 | 30.62 | 0.382 | 0.124 |
| $256 \times 256$ | CLIP | 4 | 29.87 | 0.403 | 0.116 |
| $256 \times 256$ | CLIP | 8 | 30.10 | 0.408 | 0.114 |
| $256 \times 256$ | DINOv2 | 1 | 28.18 | 0.364 | 0.130 |
| $256 \times 256$ | DINOv2 | 4 | 27.03 | 0.403 | 0.124 |
| $256 \times 256$ | DINOv2 | 8 | 27.20 | 0.400 | 0.126 |

Table 5: $k_{\mathrm{sample}}$ Ablation. FID, Precision, and Recall are computed using 10K generated samples.

$k_{\mathrm{sample}}$ depends on resolution. At $64 \times 64$, $k_{\mathrm{sample}} = 1$ gives the best FID, Precision, and Recall, indicating that a single retrieved neighbor is sufficient in low-resolution settings. At $256 \times 256$, however, moderate multi-neighbor conditioning improves fidelity. For CLIP, increasing $k_{\mathrm{sample}}$ from 1 to 4 improves FID from 30.62 to 29.87 and Precision from 0.382 to 0.403. For DINOv2, the same change improves FID from 28.18 to 27.03 and Precision from 0.364 to 0.403. These results clarify the inference-time role of the Set Transformer. When $k_{\mathrm{sample}} = 1$, it acts as a learned projection from the vision-encoder embedding space to the U-Net conditioning space. When $k_{\mathrm{sample}} > 1$, it also aggregates complementary global cues from multiple retrieved neighbors. The high-resolution results show that this aggregation can improve fidelity, confirming that the Set Transformer remains useful at inference beyond the single-neighbor case.

The ablation also reveals a fidelity–diversity trade-off. Increasing $k_{\mathrm{sample}}$ generally improves Precision but reduces Recall, especially for CLIP, indicating that stronger retrieval guidance stabilizes global structure while narrowing manifold coverage. Encoder choice affects this behavior too. DINOv2 achieves lower FID than CLIP at matched $k_{\mathrm{sample}}$ and maintains slightly higher Recall, suggesting that its visually structured embedding space preserves broader variation than CLIP's global semantic neighborhoods.

## 9 Limitation

It is important to clarify that PRISM's contribution is a significant improvement within the patch-only training paradigm. Our method does not significantly outperform baselines trained on full-resolution images. The framework's success is fundamentally tied to the retrieval set, which we construct via a simple random split. This suggests a promising direction for future work in exploring more sophisticated, diversity-aware partitioning strategies to better handle rare concepts. Furthermore, the guidance mechanism inherits any biases from the pre-trained CLIP model, and future research could investigate alternative or fine-tuned embedding models to mitigate this dependency. Finally, this approach trades reduced training computation for the increased storage and pre-processing costs required to build and index the retrieval database.

## 10 Conclusion

In this work, we addressed the critical challenge of maintaining global coherence in patch-based diffusion models without relying on full-resolution data. We introduced PRISM, a retrieval-guided framework that successfully trains high-fidelity models exclusively on image patches, significantly reducing computational overhead. Our method achieves this by retrieving semantically similar neighbors to provide external global context. We employ a Set Transformer to ensure this guidance is permutation-invariant and a dynamic annealing schedule to promote a coarse-to-fine learning process, improving both structure and detail. Our experiments validate that PRISM closes the coherence gap in patch-only methods. Most notably, our model significantly outperforms these baselines while using only a fraction of the training data, highlighting its remarkable data efficiency. This work demonstrates that retrieval-augmentation is a powerful paradigm for efficient generative modeling. Future research can build on this by exploring more advanced data partitioning strategies and extending the framework to conditional tasks like text-to-image synthesis.

## Broader Impacts

Like standard generative models, PRISM carries the risk of generating misleading content. Furthermore, relying on pre-trained encoders (e.g., CLIP) for the global semantic prior means PRISM inherently propagates their representational biases. However, because this prior is derived from an explicit retrieval database $\mathcal{D}$ rather than being entirely encoded within the diffusion U-Net's parameters, practitioners can actively audit and balance the corpus to mitigate these biases without retraining the generative backbone. Future deployments should pair PRISM with debiased encoders and rigorously audited retrieval sets to ensure equitable generation.

### Acknowledgments

SP acknowledges support from Visvesvaraya PhD fellowship. PR acknowledges support from Google Research and Satya & Rao Remala Foundation.

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

| $k_{\mathrm{sample}}$ | Decreasing | Increasing | Random | Fixed |
|---|---|---|---|---|
| 1 | 43.45 | 48.64 | 53.03 | 54.95 |
| 4 | 45.01 | 47.97 | 52.79 | 54.67 |
| 8 | 44.87 | 48.74 | 52.71 | 55.96 |
| 12 | 44.94 | 49.79 | 53.10 | 55.54 |
| 16 | 45.13 | 48.59 | 52.98 | 55.80 |
| 20 | 46.28 | 48.89 | 53.78 | 55.43 |

Table 6: FID ($\downarrow$) on CelebA with 1K training images for different scheduling strategies of $k_{\mathrm{train}}$ and $k_{\mathrm{sample}}$.

## Appendix

## A  Experiment Details

Table 7 summarizes the hyperparameters used in the training experiments for CIFAR-10, CelebA, ImageNet-100, and AFHQv2 datasets, comparing the Patch Diffusion method with our approach, PRISM. All experiments were conducted using a single GPU. The training duration was 32 million images (Mimg) for CIFAR-10, CelebA, and AFHQv2, and 96 Mimg for ImageNet-100. The minibatch size was set to 512 for CIFAR-10 and AFHQv2, 256 for CelebA, and 150 for ImageNet-100, with mixed-precision training (FP16) enabled in all cases. The learning rate was $10^{-3}$ for CIFAR-10 and AFHQv2, $10^{-4}$ for CelebA, and $2 \times 10^{-4}$ for ImageNet-100, with a ramp-up phase of 10 Mimg across all datasets. The exponential moving average (EMA) half-life was set to 0.5 Mimg for all settings. A dropout probability of 10% was used for CIFAR-10, CelebA, and AFHQv2, and 5% for ImageNet-100.

Model configurations included 64 channels for CIFAR-10 and AFHQv2 and 128 channels for CelebA and ImageNet-100, with consistent channel multipliers of $[2, 2, 2]$. The number of residual blocks per resolution was 3 for CIFAR-10 and AFHQv2, 2 for CelebA, and 4 for ImageNet-100. Attention was applied at resolutions $[8, 16]$ for CIFAR-10, AFHQv2, and ImageNet-100, and at resolution $[16]$ for CelebA. For PRISM, the maximum number of training neighbors $k_{\max}$ was set to 16 for CIFAR-10 and 32 for CelebA, ImageNet-100, and AFHQv2, and annealed to 1 according to the schedule described in Section 4.4. These consistent configurations allow a direct comparison between Patch Diffusion and PRISM across different datasets.

**AFHQv2** $256 \times 256$.  For the $256 \times 256$ resolution experiments, we modify the patch coordinate injection mechanism. Instead of concatenating the absolute $(x, y)$ patch coordinates as additional input channels, we encode them using sinusoidal positional embeddings. At higher resolutions, position information is lost if using raw scalar coordinates. Sinusoidal embeddings resolve this by providing a frequency-based, scale-invariant spatial signal. The U-Net backbone is parameterized with a base channel dimension of 96, channel multipliers of $[1, 2, 2, 2]$, and self-attention applied at the $64 \times 64$ and $32 \times 32$ feature map resolutions. We optimize the network using Adam with a learning rate of $10^{-4}$ and a dropout probability of 0.1. The model is trained for a total duration of 16M image passes using batch size of 256 (using gradient accumulation). For the retrieval-augmented conditioning, we set the maximum neighbor count to $k_{\max} = 16$ and apply a monotonically decreasing schedule throughout training.

## B  Experiments with Full-Resolution Images

Our method's largest gains appear in the patch-only regime, where the model never sees full-resolution inputs during training and the global conditioning signal must supply the missing spatial context. To understand whether retrieval guidance is still useful when the backbone has direct access to global structure, we also consider a hybrid training regime on CelebA where, for both Patch Diffusion and PRISM, 50% of the training iterations use full-resolution images and the remaining 50% use patches. Table 8 summarizes the results. When trained on $\mathcal{X}_{\mathrm{train}}$ with this hybrid schedule, Patch Diffusion achieves an FID of 10.33. Allowing Patch Diffusion to use the full dataset $\mathcal{X}$ further improves FID to 9.64, as expected from the additional supervision. Under the same hybrid scheme and using only $\mathcal{X}_{\mathrm{train}}$, PRISM attains an FID of 8.40, outperforming Patch

| Hyperparameter | CIFAR-10 | | CelebA | | ImageNet-100 | | AFHQv2 | |
| --- | --- | --- | --- | --- | --- | --- | --- | --- |
| | Patch Diffusion | PRISM | Patch Diffusion | PRISM | Patch Diffusion | PRISM | Patch Diffusion | PRISM |
| Number of GPUs | 1 | 1 | 1 | 1 | 1 | 1 | 1 | 1 |
| Duration (Mimg) | 32 | 32 | 32 | 32 | 96 | 96 | 32 | 32 |
| Minibatch size | 512 | 512 | 256 | 256 | 150 | 150 | 512 | 512 |
| Mixed-precision (FP16) | ✓ | ✓ | ✓ | ✓ | ✓ | ✓ | ✓ | ✓ |
| Learning rate $\times 10^4$ | 10 | 10 | 1 | 1 | 2 | 2 | 10 | 10 |
| LR ramp-up (Mimg) | 10 | 10 | 10 | 10 | 10 | 10 | 10 | 10 |
| EMA half-life (Mimg) | 0.5 | 0.5 | 0.5 | 0.5 | 0.5 | 0.5 | 0.5 | 0.5 |
| Dropout probability | 10% | 10% | 10% | 10% | 5% | 5% | 10% | 10% |
| Model channel | 64 | 64 | 128 | 128 | 128 | 128 | 64 | 64 |
| Channels multiplier | [2,2,2] | [2,2,2] | [2,2,2] | [2,2,2] | [2,2,2] | [2,2,2] | [2,2,2] | [2,2,2] |
| Residual blocks per resolution | 3 | 3 | 2 | 2 | 4 | 4 | 3 | 3 |
| Attention resolutions | [8,16] | [8,16] | [16] | [16] | [8,16] | [8,16] | [8,16] | [8,16] |
| $k_{\max}$ (PRISM) | – | 16 | – | 32 | – | 32 | – | 32 |

Table 7: Hyperparameters used for the training runs in Section 5. $\mathcal{X}_{\text{train}}$ and $\mathcal{D}$ follow the construction described in Section 4.1; $k_{\max}$ is used only by PRISM for dynamic neighbor scheduling.

| Method | Trained on | FID ($\downarrow$) |
| --- | --- | --- |
| Patch Diffusion | $\mathcal{X}_{\text{train}}$ | 10.33 |
| Patch Diffusion | $\mathcal{X}$ | 9.64 |
| Ours (PRISM) | $\mathcal{X}_{\text{train}}$ | **8.40** |

Table 8: FID on CelebA

Diffusion even when the latter is trained on the full dataset. This shows that retrieval-guided conditioning remains beneficial in the full-resolution setting, and can partially substitute for additional training data. At the same time, the relative improvement over the Patch Diffusion baseline is more modest than in the patch-only experiments. This behavior is expected. When the U-Net is trained exclusively on patches, it lacks global context and must infer long-range structure from limited local information. In that setting, the global conditioning from retrieved neighbors plays a dominant role in resolving ambiguities and enforcing global coherence. When full-resolution images are already seen during training, much of this global structure is directly available in the input, and the retrieval signal becomes complementary rather than essential. As a result, the gains from PRISM do not disappear but naturally saturate.

## C   Text to Image generation

While this work focuses on establishing PRISM within the unconditional generation regime, the architecture supports zero-shot text-to-image synthesis without retraining. Because the retrieval database $\mathcal{D}$ resides in the multimodal CLIP latent space, a text prompt can be encoded and directly utilized as the query vector to search $\mathcal{D}$. The retrieved top-$k$ semantic image neighbors are aggregated via the Set Transformer and passed to the pre-trained PRISM. This approach inherently bridges the CLIP modality gap. PRISM is conditioned on the image-embedding distribution it observed during training, while the semantic content is entirely driven by the user's text prompt. Formalizing and evaluating this zero-shot text-conditioned retrieval pipeline remains a primary direction for future work.

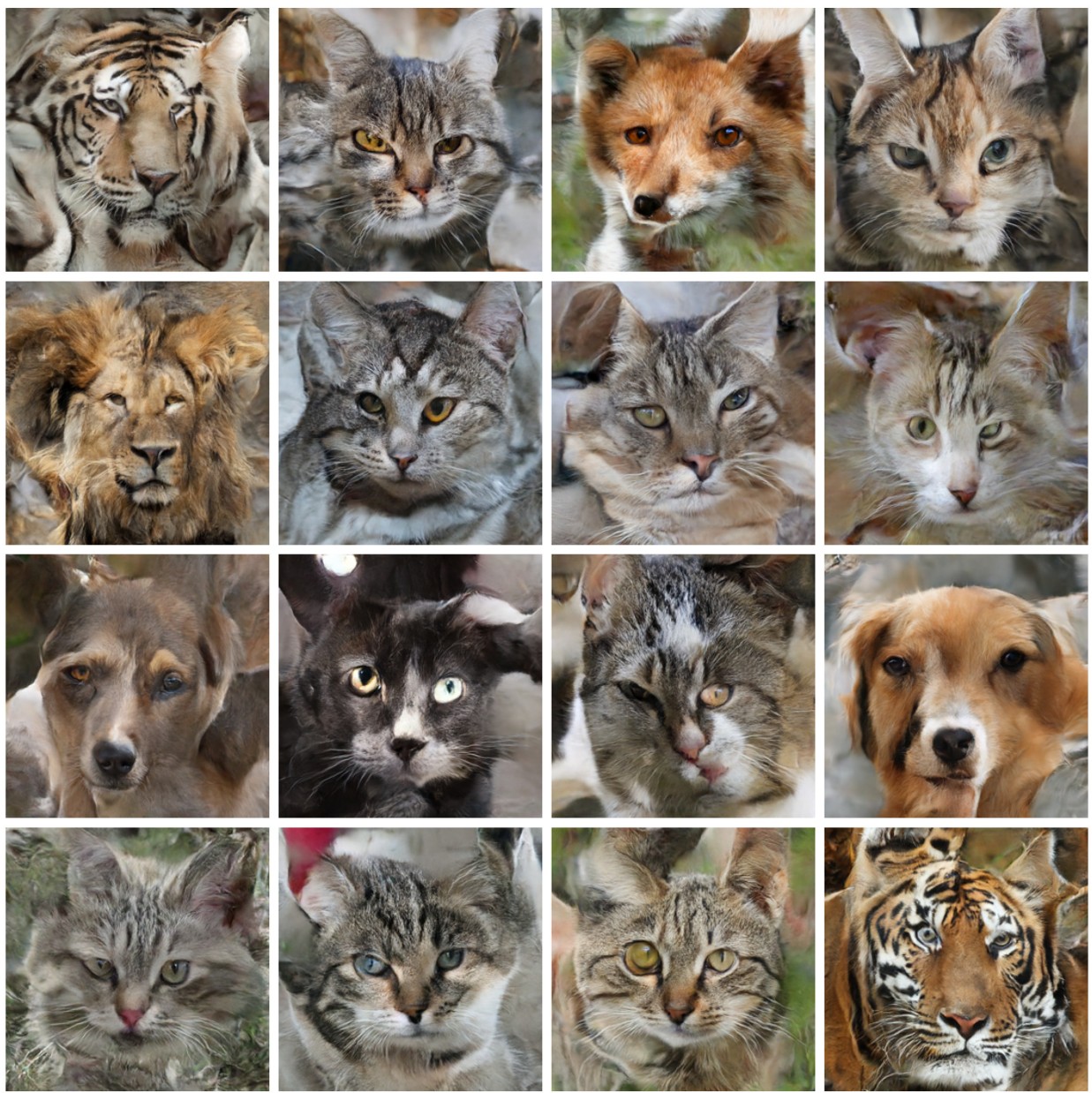

Figure 4: Images generated from PRISM AFHQv2 256x256

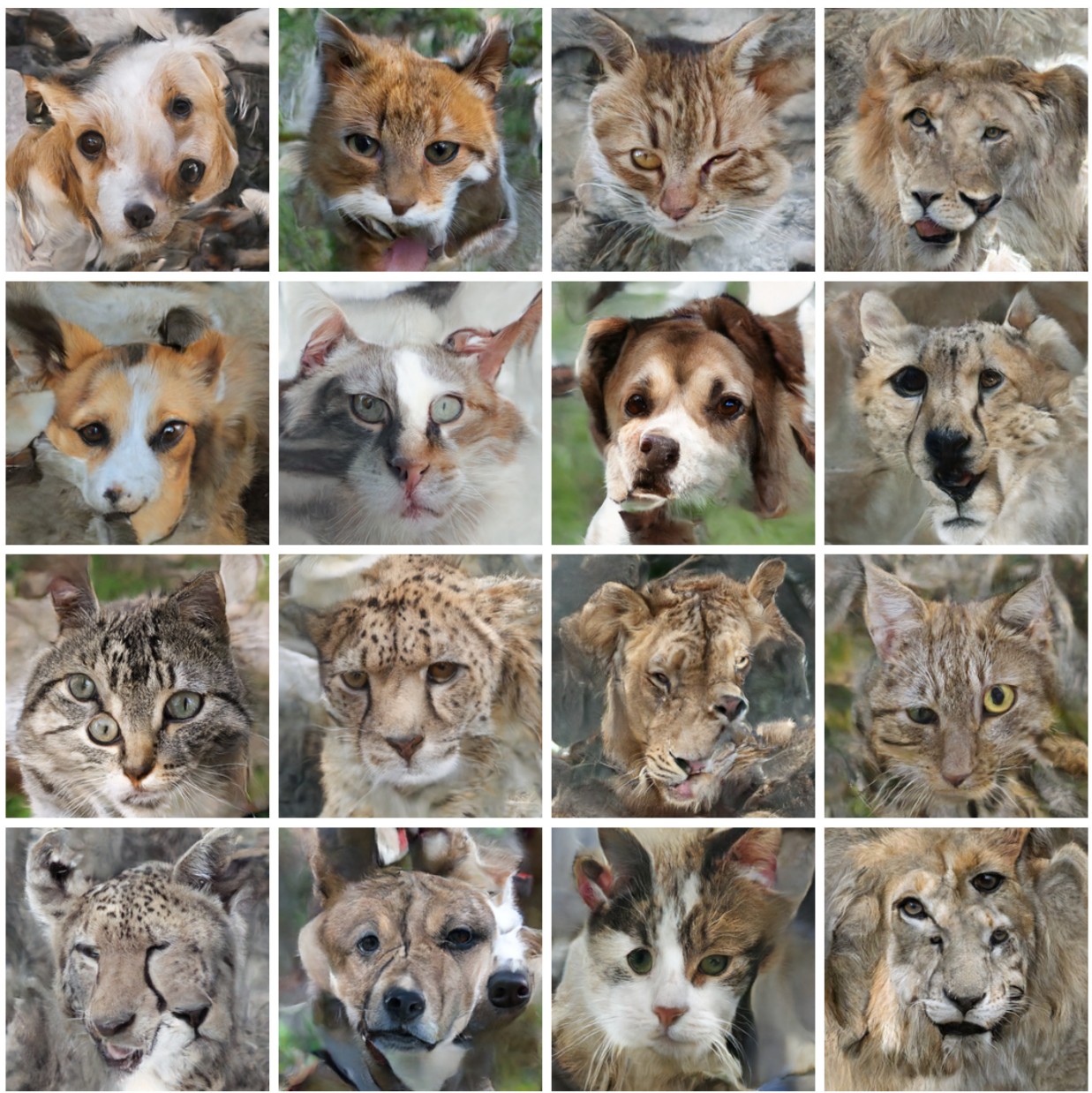

Figure 5: Images generated from Patch Diffusion AFHQv2 256x256

