# OpenReview forum: "PRISM: Patch Diffusion with Dynamic Retrieval Augmented Guidance and Permutation Invariant Conditioning"
_TMLR — Accepted by TMLR_

### Review · Reviewer_DRyH · 2025-12-13

**Summary Of Contributions:**

The paper proposes PRISM, a novel diffusion framework that is trained on image patches, with the help of RAG. In the proposed design, to make the model invariant to the order of features retrieved, a permutation-invariant set transformer is used accordingly. The authors also proposed a dynamic annealing schedule to improve training stability.

**Audience:**

Yes

**Audience Explanation:**

I think the use of a set transformer and RAG to improve coherence is the trend for more affordable computation on device side, so I think TMLR's audience may be interested.

**Claims And Evidence:**

No

**Claims Explanation:**

As pointed out by the authors, the disadvantage of training / inferencing full-size diffusion model is (A) the prohibitive GPU resource, and the problem with patch-based diffusion models is (B) incoherence.

Reading through the Method chapter, PRISM is trained on image patches, which solves problem (A), and subsequently (B) is solved with the RAG and set transformer. However, in the Experiment chapter, the authors only conduct experiment on images with low resolution, at most 64x64. This size does not pose any problem for full-size training, and I think in this sense, the claims of solving problem (A) are questionable. For problem (B), the demonstrated results suggest that introducing the conditioning can help in synthesizing coherent faces in Fig. 3, but in some cases it seems not that obvious (e.g. the results in the 2nd, 4th, 6th, and 7th columns).

**Requested Changes:**

1. Adding experiments of higher image resolution. The results on 64x64 are not persuasive. I think the full image size should at least be 512x512. This change is critical for my recommendation.

2. More about the conditioning of set transformers. Set transformer is not the contribution of this work, but using it as a condition is. I think the authors should emphasize more on how they use the set transformer (Sec. 4.3). This change will strengthen the work.

3. References. Currently the reference part is in a very rough shape. Some are missing the proceeding name. This change will strengthen the work.

---

### Review · Reviewer_qCRh · 2026-01-04

**Summary Of Contributions:**

This paper proposes PRISM, a retrieval-augmented, patch-only diffusion framework aimed at improving global coherence and data efficiency in image generation while avoiding the use of full-resolution images during training. The core idea is to compensate for the loss of global context inherent in patch-based diffusion by retrieving semantically related images using CLIP embeddings from a disjoint retrieval set and aggregating them via a permutation-invariant Set Transformer. The resulting global conditioning signal is injected into a compact EDM-DDPM++ backbone trained solely on image patches.

A key methodological contribution is the introduction of a dynamic neighbour-annealing schedule, which gradually reduces the number of retrieved neighbours during training, implementing a coarse-to-fine curriculum over contextual guidance. Extensive experiments on CIFAR-10, CelebA, ImageNet-100, and AFHQv2 demonstrate that PRISM significantly improves FID over patch-only baselines while using substantially fewer training images and comparable training time. Ablation studies further isolate the contributions of retrieval augmentation, set-based aggregation, and dynamic scheduling.

Strengths:
Addresses a well-motivated limitation of patch-only diffusion models (loss of global coherence).
Methodologically clean integration of retrieval augmentation and permutation-invariant conditioning.
Strong empirical gains across multiple datasets, especially in limited-data regimes.
Clear ablation studies supporting each design choice.

Weaknesses:
Dependence on CLIP embeddings introduces external bias and additional storage/preprocessing cost.
Improvements are primarily within the patch-only regime and do not exceed full-resolution baselines.
Evaluation is restricted to unconditional image generation tasks.

**Audience:**

Yes

**Audience Explanation:**

The findings are likely to be of interest to a broad segment of the TMLR audience, particularly researchers working on diffusion models, retrieval-augmented generation, and data- or compute-efficient learning. The paper addresses a timely and practically relevant question: how to reduce the computational and data requirements of diffusion models without sacrificing generation quality.

Moreover, the combination of patch-based training, non-parametric retrieval, and permutation-invariant conditioning is conceptually general and could inspire extensions to conditional generation, multimodal modelling, or hybrid latent/patch-based frameworks. Even for readers not directly working on image synthesis, the ideas around curriculum-guided retrieval and set-based conditioning may have broader applicability.

**Broader Impact Concerns:**

The work does not raise significant ethical concerns beyond those commonly associated with generative image models. However, the reliance on CLIP embeddings means that PRISM may inherit biases present in the pre-trained CLIP model, which could affect the diversity or fairness of generated content. A brief acknowledgment of this limitation, along with potential mitigation strategies, would be appropriate in a Broader Impact statement.

**Claims And Evidence:**

Yes

**Claims Explanation:**

The paper’s central claims that retrieval-augmented, permutation-invariant conditioning can close the coherence gap in patch-only diffusion and significantly improve data efficiency. The claims are well supported by both quantitative and qualitative evidence. Across four datasets, PRISM consistently achieves substantially lower FID scores than Patch Diffusion under identical patch-only settings, often while training on a fraction of the data and with comparable computational cost (Table 1, Section 5).

The ablation studies (Section 6) are particularly convincing: they demonstrate monotonic improvements when adding retrieval guidance, Set Transformer aggregation, and dynamic neighbour scheduling, thereby validating the necessity of each component. The analysis of different scheduling strategies further substantiates the curriculum-learning interpretation of the decreasing neighbour schedule. Qualitative results also suggest that PRISM does not simply memorize retrieved examples but instead leverages them for coherent yet novel generation. Overall, the experimental design is careful, controlled, and aligned with the paper’s claims.

**Requested Changes:**

1) Clarify retrieval-set construction and potential information leakage.
While the retrieval set is disjoint from the training set, a more explicit discussion of how this split affects generalization—and whether semantic overlap could implicitly encode label or structural information—would strengthen the methodological clarity.

2) Provide a more explicit comparison to full-resolution training efficiency.
Although the paper states that PRISM does not outperform full-resolution baselines, a concise summary of the trade-offs (quality vs. compute vs. data) would help contextualize the contribution.

3) Extend evaluation beyond unconditional generation.
Preliminary experiments or discussion on conditional tasks (e.g., class-conditional or text-conditioned generation) would broaden the impact of the work.

4) Analyze sensitivity to the retrieval encoder.
Since CLIP plays a central role, a brief ablation or discussion on alternative embedding models or fine-tuned variants would be valuable.

5) Discuss memory and scalability considerations in more detail.
Quantifying retrieval database storage and lookup costs relative to training savings would give a more complete picture of practical deployment.

---

### Review · Reviewer_HDsw · 2026-03-13

**Summary Of Contributions:**

This paper mainly focuses on the training of image diffusion
models. The authors point out the difficulty of training diffusion
models on full-sized images. They proposed to enhance the previous patch
diffusion models with RAG mechanism. Extra knowledge from a pre-built
retrieval set is retrieved during both training and inference to
guarantee the long-range consistency of generated images. Experiments
are conducted on several datasets to verify the effectiveness of the
proposed method.

**Audience:**

Yes

**Audience Explanation:**

1.  The authors provide a simple yet effective method for training
    diffusion models.

2.  In terms of the given settings, the proposed method outperforms the
    previous Patch Diffusion.

**Claims And Evidence:**

No

**Claims Explanation:**

1.  I'm mainly concerned about the timeliness of the method, which is
    only tested on image datasets with limited resolution and the very
    primitive diffusion model structure. The claim of difficulty in
    training diffusion models with full-resolution images can also be
    solved by Latent diffusion model/Latent flow matching model, which
    is more commonly adopted. In my opinion, what can really shows the
    value of the proposed method is to apply it to more recent score
    matching/flow matching models to generate high-resolution images
    (e.g. 4K or higher).

2.  The usage of set transformer is not clear enough. The authors
    mentioned using averaged or concatenated RAG embeddings leads to
    worse results, but only results regarding the averaged one are shown
    in Tab.3. Besides, I wonder why the authors have not mentioned other
    reasonable structures such as Q-Former, or averaging embeddings
    after processing them with normal transformer.

3.  It seems the proposed method is not compatible with classifier-free
    guidance, which could limit the usage in real scenarios.

4.  It would be better to provide more details about the retrieval
    process. The authors leverage CLIP's latent space for retrieval. I
    wonder if the authors could provide more comparison using other
    pretrained vision embeddings. Besides, Since the class labels are
    not engaged in the retrieval process, I wonder how to ensure the
    consistency between the class label and generated results.

5.  In Sec.4.5 the authors mentioned \$k\_{sample}=1\$ leads to the best
    results during inference. In this way, what is the role of the set
    transformer?

6.  I wonder whether the RAG mechanism would limit the variety of the
    generated results.

7.  More evaluation metrics can be included such as Precision and Recall
    to testify the coverage of the data manifold.

8. The main motivation is to solve the global coherence issue in patch-based training. However, the experiments are only conducted up to 64x64 resolution, where patches (size R/2 or R/4) already cover a large portion of the image. To truly demonstrate that PRISM captures long-range dependencies, the authors should evaluate on higher resolution images (e.g., 256x256) while maintaining small patch sizes during training.

9. Tab. 1 shows PRISM trains on 10K CIFAR-10 images in roughly 2 hours, achieving an FID of 24.68. However, there is no comparison to a standard full-resolution diffusion model (e.g., standard EDM) trained on the same 10K images for the same amount of time. Since 32x32 is already a low resolution, it is unclear whether the patch-based RAG overhead is actually more efficient than just training a standard model on that limited data.

**Requested Changes:**

Please refer to the above mentioned weaknesses. Besides, I suggest reorganizing Eq.1 for easier understanding.

---

### Decision · Action_Editor_pkCY · 2026-04-24

**Recommendation:** Accept with minor revision

**Additional Comments:**

Although this is a minor revision, this will be reviewed again. Please ensure the following two remaining concerns are formally addressed in the manuscript. Alongside your updated draft, please provide a brief summary detailing how these specific points were resolved:

* Generation Diversity Trade-offs: Explicitly acknowledge the potential reduction in diversity caused by the retrieval-augmented guidance. Please include a study measuring the degree of this issue (e.g., formalizing your Precision-Recall analysis) and provide deeper insights into these dynamics.

* Single-Image Inference Methodology: Provide a more rigorous analysis of the model's behavior during inference (specifically regarding the Set Transformer when $k=1$). We request that you:
  - Ablate the inference behavior using multiple retrieved images and report the resulting insights.
  - Analyze the role of the vision encoder in this process (e.g., expanding on the comparison between CLIP and alternatives like DINO) to clarify how the chosen embedding space influences the final generation.

**Audience:**

Yes

**Audience Explanation:**

This paper tackles an interesting and ambitious direction. The findings on using retrieval-augmented, patch-only training to reduce computational requirements will be of notable interest to TMLR's audience, particularly those working on diffusion models and data-efficient learning.

**Claims And Evidence:**

Yes

**Claims Explanation:**

This paper tackles an interesting problem: utilizing a compute-efficient patch-only training regime without sacrificing the global image coherence typically provided by full-resolution data.

The paper makes the following core claims:

1. An effective patch-based diffusion framework that achieves strong image coherence without requiring full-resolution images during training.
1. A permutation-invariant, set-based conditioning mechanism utilizing a Set Transformer.
1. A dynamic neighbor-annealing schedule ($k$) to improve training stability and generation fidelity.

Following the rebuttal phase, the reviewers agree that claims 1 and 3 are reasonably substantiated. However, one reviewer maintains a concern regarding claim 2, specifically the design and necessity of the Set Transformer during inference. Because only a single retrieved image is used at test time, its utility was questioned. The authors clarified that while the Set Transformer's primary purpose is during training, it still plays a vital role during inference by projecting the single CLIP embedding into the learned conditioning space.

Additionally, the authors acknowledge a known limitation: the retrieval mechanism may bias the model toward retrieved images, reducing generation diversity (characterized as a Precision-Recall trade-off).

The AE believes these remaining concerns are valid. However, considering the ambition of the topic and the strength of the contributions, the AE views this submission as borderline. Therefore, the AE requests a revision to afford the authors the opportunity to formally address these final issues.